# OpenReview forum: "Checkpoint-GCG: Auditing and Attacking Fine-Tuning-Based Prompt Injection Defenses"
_ICLR.cc/2026/Conference — Submitted to ICLR 2026_

### Official Review · Reviewer_zFV2 · 2025-10-27

**Soundness:** 4
**Presentation:** 4
**Contribution:** 4
**Rating:** 8
**Confidence:** 4

**Summary:**

This paper reveals the potential weakness of finetuning based defense methods against prompt injection attack. The authors proposed a method named Checkpoint-GCG which can utilize the intermediate checkpoints of safety finetuning to enhance the success of GCG method for attack. The authors show the proposed method achieves strong performance under different attack settings, indicating this method can serve as a good safety auditing method.

**Strengths:**

1. The findings and method reported in this paper is very novel to my knowledge.

2. The proposed method shows strong performance in attacking the finetuning based defense methods.

3. Different kinds of attack settings are evaluated.

4. The writing of this paper is good.

**Weaknesses:**

1. It would be better if more prompt injection attack benchmarks are used in experiments.

2. The proposed method is designed for attack finetuning based defense methods. It is not clear whether it still works when other kinds of defense methods are applied.

3. The assumption of intermediate checkpoints may be too strong for some models.

**Questions:**

Please refer to above comments.

---

> ### Author Response · Authors · 2025-11-22
>
> Thank you for your valuable feedback! We provide our responses to your comments below.
>
> > (W1) It would be better if more prompt injection attack benchmarks are used in experiments.
>
> Thank you for your suggestion!
>
> In addition to the AlpacaFarm and SEP datasets reported in Figure 4, we have conducted further experiments to evaluate the generalizability of our universal suffixes to another unseen benchmark that is potentially out-of-distribution: CyberSecEval2 [1], which contains 55 test samples.
>
> Our findings mirror existing results with the SEP dataset. The suffixes discovered by Checkpoint-GCG using 10 samples from the AlpacaFarm dataset transfer effectively, out-of-the-box, to the out-of-distribution CyberSecEval2 dataset. In contrast, suffixes discovered by standard GCG do not show any transferability, consistent with our observations on the SEP dataset. We have included these results in Section 5.3 of the revised paper.
>
> [1] Bhatt, Manish, et al. "CyberSecEval 2: A wide-ranging cybersecurity evaluation suite for large language models".
>
> > (W2) The proposed method is designed for attack finetuning based defense methods. It is not clear whether it still works when other kinds of defense methods are applied.
>
> Thank you for raising this point! Our work indeed focuses on three fine-tuning-based defenses which represent the state-of-the-art *model-level* defenses against prompt injection attacks.
>
> We agree, however, that not all defenses operate at the model level and there exist various system-level defenses. For example, detection-based defenses examine LLM prompts and responses to identify potential attacks, often using heuristics such as perplexity [1], attention scores [2], activation shifts [3], or another LLM as a detector [4, 5, 6]. For LLM agents, system-level isolation mechanisms can also limit the impact of a successful prompt injection: even if the underlying model is compromised, strict control over tool access and execution can prevent harmful actions [7, 8].
>
> However, these defenses also generally do not provide formal guarantees against all attacks. As such, a multi-layered defense strategy (defense-in-depth) is essential. Although Checkpoint-GCG does not directly target system-level defenses, it remains an important auditing and red-teaming tool for a key (model-level) layer of a multi-layer defense strategy for LLM applications.
>
> We have updated Appendix H of the paper to include this discussion - thank you for your suggestion!
>
> [1] Alon, Gabriel, et al. “Detecting Language Model Attacks with Perplexity”.
>
> [2] Hung, Kuo-Han, et al. "Attention tracker: Detecting prompt injection attacks in LLMs" Findings of the Association for Computational Linguistics: NAACL 2025. 2025.
>
> [3] Abdelnabi, Sahar, et al. “Get my drift? Catching LLM Task Drift with Activation Deltas”, 2025 IEEE Conference on Secure and Trustworthy Machine Learning (SaTML).
>
> [4] Protect AI, https://huggingface.co/protectai/deberta-v3-base-prompt-injection-v2
>
> [5] Liu, Yupei, et al. “DataSentinel: A Game-Theoretic Detection of Prompt Injection Attacks”, 2025 IEEE Symposium on Security and Privacy (SP).
>
> [6] Chennabasappa, Sahana, et al. “LlamaFirewall: An open source guardrail
> system for building secure AI agents”.
>
> [7] Debenedetti, Edoardo, et al. “Defeating Prompt Injections by Design”.
>
> [8] Beurer-Kellner, Luca, et al. “Design Patterns for Securing LLM Agents against Prompt Injections”.
>
> > (W3) The assumption of intermediate checkpoints may be too strong for some models
>
> Thank you for raising this point.
>
> We agree that access to intermediate checkpoints is a strong assumption for some models, however, such assumption is expected in auditing settings and enables practical attacks against at least some real-world deployment scenarios. For example, SecAlign [1] and SecAlign++ [2] are both open-source defense recipes, and strong models defended with SecAlign++ are already released [3], making them plausible candidates for real-world deployment. We show that Checkpoint-GCG is an effective attack in such cases, constituting a meaningful threat for at least some real-world deployments on top of the applicability of Checkpoint-GCG for auditing.
>
> [1] Chen, Sizhe, et al. “SecAlign: Defending Against Prompt Injection with Preference Optimization”, ACM CCS 2025.
>
> [2] Chen, Sizhe, et al. “Meta SecAlign: A Secure Foundation LLM Against Prompt Injection Attacks”.
>
> [3] https://huggingface.co/facebook/Meta-SecAlign-8B
>
> **We hope our response helps provide clarity and address your concerns. Please let us know if there are any remaining issues, suggestions, or questions.**

---

> > ### Comment · Reviewer_zFV2 · 2025-11-24
> >
> > Thank you very much for your response. My questions are well addressed.

---

### Official Review · Reviewer_2iXw · 2025-10-29

**Soundness:** 3
**Presentation:** 3
**Contribution:** 2
**Rating:** 4
**Confidence:** 4

**Summary:**

This paper introduces Checkpoint-GCG, an extension of the original GCG attack designed to audit and attack fine-tuning based defenses against prompt injection (e.g., StruQ, SecAlign, SecAlign++). The key insight is that intermediate fine-tuning checkpoints can serve as stepping stones to gradually optimize adversarial suffixes that remain effective even as defenses strengthen. The core idea is to leverage intermediate fine-tuning checkpoints as stepping stones that guide the optimization of adversarial suffixes, enabling the attack to remain effective as the model becomes more robust. The authors demonstrate that Checkpoint-GCG substantially outperforms standard GCG, achieving much higher attack success rates. Moreover, the optimized adversarial suffixes can be generalized to unseen prompts and can transfer to related models within the same family.

**Strengths:**

1. The idea of leveraging intermediate fine-tuning checkpoints as initialization stages for GCG is largely novel and well-motivated, bridging a gap between training dynamics and attack optimization.
2. The experiments are sufficient, as the authors evaluate multiple model families (Llama-3-8B, Mistral-7B, Qwen2-1.5B), several defense mechanisms (StruQ, SecAlign, SecAlign++), and multiple threat settings (auditing, universal attack, model transferability).
3. The performance of the attacks are impressive: for example, Checkpoint-GCG achieves 96% attack success rate against SecAlign, compared to only 6% for standard GCG.

**Weaknesses:**

1. Effectiveness due to additional information: The strong performance of Checkpoint-GCG is not entirely surprising, as it benefits from substantially more information (specifically, access to intermediate fine-tuning checkpoints as part of a defense mechanism) than standard attacks, which gives it an inherent advantage. Throughout the paper, the authors primarily compare Checkpoint-GCG against the standard GCG baseline to emphasize its improvements; while this comparison is acceptable, it may inadvertently mislead readers who overlook the fact that the proposed method operates under a significantly stronger information assumption. For example, the authors limit the number of GCG steps to ensure a “fair” comparison, but this fairness is not very meaningful since Checkpoint-GCG already benefits from access to more information.
2. The threat model is unrealistic: The assumed access to intermediate checkpoints and full input context is unrealistic in practical deployment settings. Although the authors acknowledge this limitation and attempt to relax these assumptions, the empirical results show that the attacks fail to transfer effectively when models come from different families, limiting real-world applicability. In line 431, the authors also claim that “it is still realistic in practice, as organizations may open-source a model or defense before deploying an update behind an API”. However, this justification remains speculative and does not reflect most real-world deployment scenarios, where fine-tuning checkpoints from defensive retraining are rarely made publicly available, yet Checkpoint-GCG critically depends on access to such checkpoints from a sufficiently similar surrogate model.
3. Limited theoretical insight: While the method demonstrates strong empirical results, it lacks a formal analysis or theoretical explanation for why progressing through fine-tuning checkpoints leads to better optimization outcomes. Including such insights, even at a conceptual level, could significantly strengthen the paper’s contribution beyond empirical findings.

**Questions:**

1. Since this work demonstrates the effectiveness of Checkpoint-GCG, what would be some effective defense mechanisms against it? The authors could consider adding some relevant discussions to the paper.
2. The authors evaluate four different strategies for selecting checkpoints, but the total number of checkpoints used in the main experiments remains relatively large (around 100). Could the number of checkpoints be further reduced to improve computational efficiency? It would also be helpful to analyze how sensitive the attack performance is to this trade-off between the number of checkpoints and overall effectiveness.
3. When computing success@k, how are the k independent attack attempts generated or launched?

---

> ### Author Response · Authors · 2025-11-22
>
> Thank you for your valuable feedback! We provide our responses to your comments and questions below.
> > (W1) Effectiveness due to additional information: The strong performance of Checkpoint-GCG is not entirely surprising, as it benefits from substantially more information (specifically, access to intermediate fine-tuning checkpoints as part of a defense mechanism) than standard attacks, which gives it an inherent advantage. Throughout the paper, the authors primarily compare Checkpoint-GCG against the standard GCG baseline to emphasize its improvements; while this comparison is acceptable, it may inadvertently mislead readers who overlook the fact that the proposed method operates under a significantly stronger information assumption. For example, the authors limit the number of GCG steps to ensure a “fair” comparison, but this fairness is not very meaningful since Checkpoint-GCG already benefits from access to more information.
>
> In the auditing threat model, Checkpoint-GCG indeed benefits from access to intermediate fine-tuning checkpoints, whereas standard GCG does not. We agree that our phrasing of “fair comparison” in Section 4 could have inadvertently suggested otherwise - thank you for pointing this out! We removed this subsentence “To ensure a fair comparison with Checkpoint-GCG” from the “Baselines” paragraph of Section 4 to avoid any implication that Checkpoint-GCG and standard GCG assume access to the same level of information, and have added a clarifying sentence in Section 5.2 explicitly stating that since our evaluation here is conducted in an auditing setting, access to intermediate checkpoints is feasible and expected, and the goal is to determine whether successful attacks exist.
>
> We want to clarify, however, that simply having access to more information, does not necessarily translate to better performances. For instance, in privacy, white-box membership inference attacks do not outperform black-box loss-based ones [1].
>
> [1] Cretu, Ana-Maria, et al. “Investigating the Effect of Misalignment on Membership Privacy in the White-box Setting”, Proceedings on Privacy Enhancing Technologies (PoPETs 2024). Link: https://arxiv.org/abs/2306.05093
>
> > (W2) The threat model is unrealistic: The assumed access to intermediate checkpoints and full input context is unrealistic in practical deployment settings. Although the authors acknowledge this limitation and attempt to relax these assumptions, the empirical results show that the attacks fail to transfer effectively when models come from different families, limiting real-world applicability. In line 431, the authors also claim that “it is still realistic in practice, as organizations may open-source a model or defense before deploying an update behind an API”. However, this justification remains speculative and does not reflect most real-world deployment scenarios, where fine-tuning checkpoints from defensive retraining are rarely made publicly available, yet Checkpoint-GCG critically depends on access to such checkpoints from a sufficiently similar surrogate model.
>
> Indeed, on top of our auditability results, we also study the transferability of suffixes to other models. We agree that while these results show good transferability to models of the same family, our suffixes do not transfer across model families.
>
> Given the rapid growth of open-source, on-device, on premises model releases and the release of defense training recipes including by leading model providers, we think that transferability across models from the same family is an important risk to be aware of. For example, SecAlign [1] and SecAlign++ [2] are both open-source defense recipes, and strong models defended with SecAlign++ are already released [3], making them plausible candidates for real-world deployment. We show that Checkpoint-GCG is an effective attack in such cases, constituting a meaningful threat for at least some real-world deployments on top of the applicability of Checkpoint-GCG for auditing.
>
> [1] Chen, Sizhe, et al. “SecAlign: Defending Against Prompt Injection with Preference Optimization”, ACM CCS 2025. Link: https://arxiv.org/abs/2410.05451
>
> [2] Chen, Sizhe, et al. “Meta SecAlign: A Secure Foundation LLM Against Prompt Injection Attacks”. Link: https://arxiv.org/abs/2507.02735
>
> [3] https://huggingface.co/facebook/Meta-SecAlign-8B

---

> > ### Author Response · Authors · 2025-11-22
> >
> > > (W3) Limited theoretical insight: While the method demonstrates strong empirical results, it lacks a formal analysis or theoretical explanation for why progressing through fine-tuning checkpoints leads to better optimization outcomes. Including such insights, even at a conceptual level, could significantly strengthen the paper’s contribution beyond empirical findings.
> >
> > Thank you for the suggestion! We completely agree that further insights for why Checkpoint-GCG works would be valuable. Below, we first provide a theoretical intuition for why progressing through checkpoints works, given the defense’s fine-tuning objective and Checkpoint-GCG’s optimization objective. Then, we also provide results from a new interpretability experiment based on attention scores that provides further intuition for why Checkpoint-GCG works.
> >
> > **Theoretical intuition:**
> >
> > The attacker seeks to maximize $f(s; \theta) = \log P_\theta(y^* \mid p \mid \mid s)$ over suffixes $s \in \mathcal{V}^l$ (also see Sec. 2). During fine-tuning, the model parameters are updated to minimize a fine-tuning loss $\ell_{\text{fine-tuning}}(\theta)$ that penalizes undesirable completions (e.g., via DPO), moving from an initial checkpoint $\theta_0$ (undefended) to a final model $\theta_C$.
> >
> > Because this fine-tuning objective discourages the model from predicting undesirable completions, fine-tuning updates are expected to also reduce $f$. If we update the model parameters from $\theta_{c}$ to $\theta_{c+1}$, a first order approximation in the change of $f$ could be written as:
> >
> > $$
> > f(s; \theta_{c+1}) \approx f(s; \theta_c) - \eta\ \nabla_\theta f(s; \theta_c)^\top \nabla_\theta \ell_{\text{fine-tuning}}(\theta_c).
> > $$
> >
> > where $\eta$ is the learning rate. If the gradients $\nabla_\theta f(s; \theta_c)$ and $ \nabla_\theta \ell_{\text{fine-tuning}}(\theta_c)$ are aligned (which is likely the case as the fine-tuning loss is explicitly designed to steer away from harmful outputs), then the fine-tuning process reduces the model's likelihood of generating the harmful target $y^*$ given $p  \mid \mid s$. Intuitively, thus, as fine-tuning progresses, $f(s; \theta)$ becomes smaller, and it likely becomes harder to find the optimal suffix $\arg \max_s f(s; \theta)$.
> >
> > Further, if we assume that the optimal suffix $s_{\theta, max} \in \arg \max_s f(s; \theta)$ varies continuously with $\theta$, *warm-starting* of the suffix optimization using intermediate checkpoints would intuitively help. Indeed, initializing the optimization at checkpoint $\theta_{c+1}$ with $s^{(c)} \approx s_{\theta_c, max}$ keeps the search near $s_{\theta_{c+1}, max}$ (within the basin of attraction), accelerating convergence. We have now added this intuition to Appendix B in the revised paper.
> >
> > **Interpretability experiment:**
> >
> > Inspired by recent work that analyzes changes in model activations and attention patterns to understand and detect prompt injection attacks [1,2], we now analyze how the model’s attention patterns shift with Checkpoint-GCG suffixes.
> >
> > Specifically, we compute the final model’s attention score of user prompt ($Attn_{U}$) and its attention score of injected instruction + suffix ($\text{Attn}_{A}$), with suffixes obtained at each checkpoint during Checkpoint-GCG. We plot these attention scores across all samples where Checkpoint-GCG against the final model is successful in Figure 5a (see Section 5.4 for experimental details).
> >
> > We find that these suffixes, obtained during Checkpoint-GCG’s optimization, smoothly and monotonously shift the final model’s attention away from the user prompt towards the adversarial injection. Although these suffixes are optimized against intermediate checkpoints and not the final model, they still progressively steer the final model’s attention towards the adversarial injection – indicating that the greedy optimization of Checkpoint-GCG at each checkpoint gradually steers the suffix towards a successful attack. Notably, these attention scores evolve in a remarkably smooth and monotonous manner, especially as Checkpoint-GCG does not directly optimize for attention scores. We then also show that, for checkpoints where these attention scores shift the most, Checkpoint-GCG spends most of its budget (Figure 5b).
> >
> > Taken together, we really appreciate your suggestion, and believe this experiment and its insights further improve our work. We have now added this experiment and discussion to Section 5.4.
> >
> > [1] Hung, Kuo-Han, et al. "Attention tracker: Detecting prompt injection attacks in LLMs" Findings of the Association for Computational Linguistics: NAACL 2025. 2025. Link: https://arxiv.org/abs/2411.00348
> >
> > [2] Abdelnabi, Sahar, et al. “Get my drift? Catching LLM Task Drift with Activation Deltas”, 2025 IEEE Conference on Secure and Trustworthy Machine Learning (SaTML). Link: https://arxiv.org/abs/2406.00799

---

> > > ### Author Response · Authors · 2025-11-22
> > >
> > > > (Q1) Since this work demonstrates the effectiveness of Checkpoint-GCG, what would be some effective defense mechanisms against it? The authors could consider adding some relevant discussions to the paper.
> > >
> > > Thank you for the suggestion to add a discussion of potential defenses in the paper! We have included the following to Appendix H in the revised version:
> > >
> > > Although fine-tuning-based defenses will likely continue to improve, training models to be fully robust to prompt injections is likely to remain a challenging task in the near future. Future defenses are therefore unlikely to fully protect or provide guarantees against all attacks, much like adversarial examples in the domain of computer vision, which remain an unsolved problem despite more than a decade of research. As such, a multi-layered defense strategy (defense-in-depth) is essential. For LLMs, detection systems can be deployed on top of models fine-tuned with prompt injection defenses. These systems examine prompts and responses to identify potential attacks, often using heuristics such as perplexity [1], attention scores [2], activation shifts [3], or another LLM as a detector [4, 5, 6]. For LLM agents, system-level isolation mechanisms can further limit the impact of a successful prompt injection: even if the underlying model is compromised, strict control over tool access and execution can prevent harmful actions [7, 8].
> > >
> > > Finally, Checkpoint-GCG itself can also be incorporated into the development of stronger prompt injection defenses; we provide preliminary experiments and insights toward this in Appendix H.
> > >
> > >
> > > [1] Alon, Gabriel, et al. “Detecting Language Model Attacks with Perplexity”, https://arxiv.org/abs/2308.14132
> > >
> > > [2] Hung, Kuo-Han, et al. "Attention tracker: Detecting prompt injection attacks in LLMs" Findings of the Association for Computational Linguistics: NAACL 2025. 2025. Link: https://arxiv.org/abs/2411.00348
> > >
> > > [3] Abdelnabi, Sahar, et al. “Get my drift? Catching LLM Task Drift with Activation Deltas”, 2025 IEEE Conference on Secure and Trustworthy Machine Learning (SaTML). Link: https://arxiv.org/abs/2406.00799
> > >
> > > [4] Protect AI, https://huggingface.co/protectai/deberta-v3-base-prompt-injection-v2
> > >
> > > [5] Liu, Yupei, et al. “DataSentinel: A Game-Theoretic Detection of Prompt Injection Attacks”, 2025 IEEE Symposium on Security and Privacy (SP), Link: https://arxiv.org/abs/2504.11358
> > >
> > > [6] Chennabasappa, Sahana, et al. “LlamaFirewall: An open source guardrail system for building secure AI agents”. Link: https://arxiv.org/abs/2505.03574
> > >
> > > [7] Debenedetti, Edoardo, et al. “Defeating Prompt Injections by Design”. Link: https://arxiv.org/abs/2503.18813
> > >
> > > [8] Beurer-Kellner, Luca, et al. “Design Patterns for Securing LLM Agents against Prompt Injections”. Link: https://arxiv.org/abs/2506.08837

---

> > > > ### Author Response · Authors · 2025-11-22
> > > >
> > > > > (Q2) The authors evaluate four different strategies for selecting checkpoints, but the total number of checkpoints used in the main experiments remains relatively large (around 100). Could the number of checkpoints be further reduced to improve computational efficiency? It would also be helpful to analyze how sensitive the attack performance is to this trade-off between the number of checkpoints and overall effectiveness.
> > > >
> > > > Thank you for raising this point. Below, we further reduce the number of selected checkpoints by increasing the $\tau_\text{grad}$ threshold in the GRAD checkpoint selection strategy, and show that attack effectiveness can be maintained with substantially fewer checkpoints (e.g., 29 checkpoints with $\tau_\text{grad}=0.3$), while also illustrating a trade-off between number of checkpoints and per-checkpoint GCG steps.
> > > >
> > > > In Table 4 of the submission, we compare multiple checkpoint selection strategies with varying hyperparameters on a subset of 20 samples against the most robust model that we study (SecAlign-defended Llama-3-8B-Instruct).
> > > >
> > > > To further examine the relationship between number of selected checkpoints and attack effectiveness, we ran additional experiments with larger $\tau_\text{grad}$ values for our selected checkpoint selection strategy, GRAD, to reduce the number of selected checkpoints. Results for GRAD across all tested $\tau_\text{grad}$ values are reported below.
> > > >
> > > > | Checkpoint Strategy | Parameter values | ASR % ↑ | # Selected checkpoints ↓ | Total Checkpoint-GCG steps (avg across samples) ↓ |
> > > > |--|--|--|--|--|
> > > > |grad|$\tau_{\text{grad}}=0.05$| 100 | 102 | 3077 |
> > > > ||$\tau_{\text{grad}}=0.1$ | 100 | 64 | 2033 |
> > > > ||$\tau_{\text{grad}}=0.2$ | 90 | 41 | 1764 |
> > > > ||$\tau_{\text{grad}}=0.3$ (new) | 90 | 29 | 1475 |
> > > > ||$\tau_{\text{grad}}=0.4$ (new) | 50 | 19 | 1721 |
> > > > ||$\tau_{\text{grad}}=0.5$ (new) | 45 | 12 | 1213 |
> > > > ||$\tau_{\text{grad}}=0.6$ (new) | 20 | 5 | 798 |
> > > >
> > > >
> > > > These results show a trade-off between the number of selected checkpoints and the average number of GCG steps required at each checkpoint. Lowering $\tau_\text{grad}$ increases the number of selected checkpoints, but the number of per-checkpoint GCG steps decreases accordingly. As a result, *the total Checkpoint-GCG cost does not increase substantially* with a higher number of selected checkpoints ($\tau_\text{grad}=0.3$ vs $\tau_\text{grad}=0.5$), while having more selected checkpoints allows for smoother optimization toward the final checkpoint and higher ASRs. We visualize this trade-off in a new plot (Figure 3b) and discuss the results in Section 5.2 and Appendix C.
> > > >
> > > >
> > > > Finally, although the total number of Checkpoint-GCG steps is typically 2-4x the standard GCG budget of $T=500$, these additional steps yield substantial improvements in attack effectiveness. With roughly 3x the standard GCG budget, Checkpoint-GCG achieves ASRs above 90%, whereas standard GCG achieves only 6% (Table 1). In an auditing setting and at least some high-risk attack settings (e.g., zero-day attacks, attacks against agentic payment systems), we believe this trade-off is justified: a modest increase in compute significantly improves the chances of discovering vulnerabilities that standard methods miss.
> > > >
> > > > > (Q3) When computing success@k, how are the k independent attack attempts generated or launched?
> > > >
> > > > Thank you for the clarifying question!
> > > >
> > > > success@k measures the probability of achieving at least one successful attack within k independent attack attempts. For each defended model, we apply the same universal suffix to all independent test samples and compute the empirical attack success rate (ASR), which is the fraction of samples on which the attack succeeds.
> > > >
> > > > Since these test samples are independent, success@k can be computed as $1 - (1-\text{ASR})^k$, which represents the probability that an attacker using the same universal suffix succeeds on at least one of the k target samples (i.e., succeeds in at least one attack attempt).
> > > >
> > > > **We hope our responses and additional experiments adequately address your concerns and provide further clarity. If so, we would be grateful if you might consider updating your assessment of the paper.**

---

### Official Review · Reviewer_62dP · 2025-10-30

**Soundness:** 2
**Presentation:** 3
**Contribution:** 1
**Rating:** 2
**Confidence:** 4

**Summary:**

This paper introduces Checkpoint-GCG, a method that leverages intermediate checkpoints of a fine-tuned model to enhance the effectiveness of GCG, a white-box adversarial attack. The method shows substantial improvements over vanilla GCG and is posed as an auditing tool for prompt injection and jailbreaking defenses. Further, the adversarial strings found by the method are transferable to other models.

**Strengths:**

1. The method serves as a useful tool for improving model auditing, since intermediate checkpoints will be available to model providers.
2. The evaluation considers strong baselines for comparison against state-of-the-art defenses and shows that the attack is highly effective and transferable.

**Weaknesses:**

A nearly identical approach for jailbreaking has already been published at ICLR 2025. Wang et al. [1] introduced a staged jailbreaking technique that converts a challenging optimization problem (i.e., jailbreaking an aligned model with GCG) into a sequence of easy-to-hard problems, where the solution of each prior problem is used to warm-start the optimization of the next problem. Here, each problem is a model checkpoint, obtained by deliberately misaligning the model, making it easier to attack. While the checkpoint selection schemes, method for obtaining them and optimization targets are different, the core methodology is the same.
___
### References
[1] Functional Homotopy: Smoothing Discrete Optimization via Continuous Parameters for LLM Jailbreak Attacks; Wang et al. ICLR 2025. Link: https://arxiv.org/abs/2410.04234

**Questions:**

1. For the completeness of evaluation, how does Checkpoint-GCG perform with a total budget of 500 steps? Is there a threshold beyond which it becomes more viable than vanilla GCG?
2. What does the distribution of Checkpoint-GCG budgets look like for successful attacks? For attacks where vanilla GCG is successful, how does Checkpoint-GCG fare?

---

> ### Author Response · Authors · 2025-11-22
>
> Thank you for your valuable feedback! We provide our responses to your comments and questions below.
>
> > (W) Overlap with Wang et al.
>
> Thank you very much for pointing us to Wang et al.’s Functional Homotopy (FH) paper. We conducted a continuous and thorough literature review, including examining hundreds of papers that cite the original GCG work by Zou et al. We were aware of prior easy-to-hard approaches in adversarial attacks, including [1, 2, 3, 4], and attacks leveraging intermediate checkpoints [5, 6]. Despite this effort, the FH paper did not surface in our search, and we genuinely appreciate you bringing it to our attention. We have added an appropriate citation in the revised version of the paper (Section 6).
>
> Overall, we agree that the idea of easy-to-hard optimization in adversarial contexts is not new. For example, Samvelyan et al. [1] uses increasingly high-performing jailbreak prompts as stepping stones; Cai et al. [2] and He et al. [3] propose training models against progressively stronger attacks to achieve better adversarial robustness; and Jia et al. [4] also propose an easy-to-hard variant of GCG: initializing GCG on difficult prompts using suffixes that succeed on easier ones (note that we compare Checkpoint-GCG to Jia et al. [4] in Appendix F and show Checkpoint-GCG to be substantially more effective). Similarly, we agree that previous work used intermediate model checkpoints to strengthen attacks: Li et al. [5] demonstrate this for highly transferable adversarial attacks in the computer vision domain, and Liu et al. [6] exploit full training trajectories for membership inference attacks.
>
> After carefully studying the FH paper, we agree that it is very nice work and a compelling jailbreak attack that creates checkpoints by deliberately misaligning safety-aligned chat models. We, however, strongly believe that our work complements previous work in the literature, including the FH paper, from which it differs in several ways:
> - First, while both papers rely on an easy-to-hard approach, our paper focuses on *auditing models specifically defended against prompt injection* while the FH paper misaligns models to enable jailbreaks attacks. Our work proposes an *auditing method based on actual model checkpoints* (which are available for free to the model developer) and uses it to examine three different state-of-the-art (SOTA) defenses against prompt injection attacks.
> - Second, we show that *universal attacks exist against SOTA prompt injection defenses*. The universality of the attack is indeed crucial for prompt injections to be a risk in practice, as neither the system prompt, nor the user task, for which the injected document is retrieved is known. Standard GCG does indeed sometimes succeed at finding non-universal attacks, but it fails to find universal ones. Universality for jailbreaks is limited to only system prompts and, from the FH paper, it’s unclear whether this would be feasible using misaligned models.
> - Third, we propose a variety of checkpoint selection strategies and provide insights into the synergy between training dynamics and attack effectiveness, showing that the *final fine-tuned model’s attention* increasingly shifts to the injected instruction and adversarial suffix with suffixes *obtained at intermediate checkpoints* (Section 5.4). We now also implement the binary search strategy used in the FH paper and show it to take roughly 3 times as many GCG steps as our approach (Appendix C).
> - Fourth, while our focus is on model auditing, we also show our attack to transfer black-box across models of the same family even if they are fine-tuned with a different prompt injection defense, providing important implications for model developers.
> - Finally, we now also include a new section showing that including Checkpoint-GCG suffixes in the fine-tuning data could make the defense more robust (Appendix H).
>
> [1] Samvelyan, Mikayel, et al. "Rainbow teaming: Open-ended generation of diverse adversarial prompts." Advances in Neural Information Processing Systems 37 (2024).
>
> [2] Cai, Qi-Zhi, et al. "Curriculum adversarial training." Proceedings of the Twenty-Seventh International Joint Conference on Artificial Intelligence (IJCAI-18).
>
> [3] He, Lirong, et al. "Boosting adversarial robustness via self-paced adversarial training." Neural Networks 167 (2023).
>
> [4] Jia, Xiaojun, et al. “Improved Techniques for Optimization-Based Jailbreaking on Large Language Models”, ICLR 2025.
>
> [5] Li, Shixin, et al. “Enhancing Adversarial Transferability with Checkpoints of a Single Model’s Training”, 2025 IEEE/CVF Conference on Computer Vision and Pattern Recognition (CVPR).
>
> [6] Liu, Yiyong, et al. “Membership Inference Attacks by Exploiting Loss Trajectory”, 2022 ACM SIGSAC Conference on Computer and Communications Security.

---

> > ### Author Response · Authors · 2025-11-22
> >
> > > (W) Overlap with Wang et al. (CONTINUED)
> >
> > **New experiments:**
> >
> > We implemented the binary search strategy for selecting intermediate checkpoints proposed in the FH paper. We find that it achieved a 100% ASR against SecAlign-defended Llama-3-8B-Instruct using only 19.35 +- 9.79 checkpoints (mean +- standard deviation across 20 samples), however, it required an average of 6215 total GCG steps per sample. For context, this is roughly 3x the total number of steps per-sample required by Checkpoint-GCG using our GRAD checkpoint selection strategy (Table 4). We now also include Figure 3b, which further clarifies the ASR and trade-offs for various checkpoint selection strategies.
> >
> > Fine-tuning with Checkpoint-GCG suffixes: We have now also added new experiment results in Appendix H, which provide preliminary insights into how Checkpoint-GCG can be integrated into prompt injection defense development: incorporating suffixes discovered by Checkpoint-GCG into the fine-tuning data further improves robustness, whereas incorporating suffixes found by standard GCG yields little improvement.
> >
> > Overall, while we acknowledge that previous work exists, including the FH paper which we now reference, we believe that the community would greatly benefit from this work, including as an auditing method to gain a clearer understanding of the robustness of state-of-the-art prompt injection defenses, and to further understand the success and risks associated with universal and transferable attacks.

---

> > > ### Author Response · Authors · 2025-11-22
> > >
> > > > (Q1) For the completeness of evaluation, how does Checkpoint-GCG perform with a total budget of 500 steps? Is there a threshold beyond which it becomes more viable than vanilla GCG?
> > >
> > > Our focus on this work is really on auditing, where we believe a slightly higher cost is reasonable to test the vulnerability of a model before deploying it, especially compared to the fine-tuning cost.
> > >
> > > However, we agree that studying the effectiveness of the method under smaller budgets is valuable. In our experiments, we do not directly set a total budget, but instead rely on setting $\tau_{grad}$ thresholds for selecting checkpoints, which then affects the total GCG steps taken across all selected checkpoints. Varying $\tau_{grad}$, we find that reducing the total optimization budget across all checkpoints to fewer than 1000 steps leads to a substantial drop in Checkpoint-GCG’s performance, down to roughly a 20% ASR. Although this still outperforms standard GCG, we believe that allocating a slightly larger budget, for example, 1475 steps for $\tau_{grad}=0.3$ , which yields a 90% ASR, is a worthwhile trade-off. These results have been added to Table 4.
> > >
> > > Additionally, we observe a trade-off between the number of selected checkpoints and the average number of GCG steps required at each checkpoint. Lowering $\tau_\text{grad}$ increases the number of selected checkpoints, but the number of per-checkpoint GCG steps decreases accordingly. As a result, the total Checkpoint-GCG cost does not increase substantially, while having more selected checkpoints allows for smoother optimization toward the final checkpoint and higher ASRs. We illustrate this trade-off in a new plot (Figure 3b) and discuss the results in Section 5.2 and Appendix C.
> > >
> > >
> > > > (Q2) What does the distribution of Checkpoint-GCG budgets look like for successful attacks? For attacks where vanilla GCG is successful, how does Checkpoint-GCG fare?
> > >
> > > We have now included Appendix D discussing the distribution of Checkpoint-GCG steps for the successful attacks. While for some samples, Checkpoint-GCG requires a few thousand steps, 45% of the successful samples need less than 1000 total steps across all selected checkpoints.
> > >
> > > In contrast, allowing standard GCG to run for as many steps as Checkpoint-GCG required for each sample (Checkpoint-GCG budget) does not help increase its attack success rate: it can spend thousands of steps per sample, but still fails to achieve a successful attack for most samples.
> > >
> > > **We hope our responses and additional experiments adequately address your concerns and provide further clarity. If so, we would be grateful if you might consider updating your assessment of the paper.**

---

### Official Review · Reviewer_HERg · 2025-10-31

**Soundness:** 3
**Presentation:** 4
**Contribution:** 4
**Rating:** 4
**Confidence:** 4

**Summary:**

This paper introduces Checkpoint-GCG, a white-box adversarial attack method designed to audit and bypass fine-tuning-based prompt injection defenses for large language models (LLMs). Prompt injection attacks exploit LLMs’ inability to distinguish trusted instructions from malicious content in untrusted data, while existing fine-tuning defenses aim to mitigate this by training models. Meanwhile, traditional attacks like Greedy Coordinate Gradient (GCG) suffer from drastically reduced attack success rates (ASR) against these defenses. To address this, Checkpoint-GCG leverages intermediate fine-tuning checkpoints as "stepping stones": it sequentially optimizes adversarial suffixes across checkpoints, using the suffix from the previous checkpoint to initialize the next, thereby improving attack effectiveness. The experiment results show that Checkpoint-GCG successfully improves the attack performance.

**Strengths:**

- The paper is easy to read and well-written.


- Checkpoint-GCG addresses a critical gap in LLM security: the failure of traditional attacks to evaluate the robustness of state-of-the-art fine-tuning defenses. By exploiting the parameter updates in fine-tuning checkpoints, it provides a principled solution to GCG’s initialization sensitivity.

**Weaknesses:**

- The auditing setting (full access to model checkpoints and the exact input) is realistic for internal red-teaming but not for many real-world attackers. The authors do relax these assumptions, but the highest ASRs require checkpoint access. The practical feasibility of obtaining intermediate checkpoints for deployed proprietary models is limited.

- Checkpoint-GCG runs GCG many times across selected checkpoints; while GRAD reduces cost, the totals in reported experiments are nontrivial (per-sample totals reported in Table 4). The method uses large per-checkpoint budgets (authors set T up to 1000 with early stopping), which may be computationally expensive in practice for large models.

- The paper lacks systematic interpretability experiments or mechanistic explanations for why Checkpoint-GCG achieves superior attack performance.

**Questions:**

Listed in Weakness.

---

> ### Author Response · Authors · 2025-11-22
>
> Thank you for your valuable feedback! We provide our responses to your comments and questions below.
>
> > (W1) The auditing setting (full access to model checkpoints and the exact input) is realistic for internal red-teaming but not for many real-world attackers. The authors do relax these assumptions, but the highest ASRs require checkpoint access. The practical feasibility of obtaining intermediate checkpoints for deployed proprietary models is limited.
>
> Thank you for raising this point. We believe model auditing is important, especially for high-risk prompt injection vulnerabilities and for measuring progress in proposed defenses. However, we also show that Checkpoint-GCG remains notably effective even without access to checkpoints or the exact input, achieving over 60% ASR in a fully black-box setting against a state-of-the-art defended model, which we believe offers meaningful insight since such defended models may well be deployed in real-world systems. We elaborate further below.
>
> **Auditing.** Our primary focus is indeed on auditing. We agree that access to intermediate checkpoints and full input prompts is not representative of many real-world attacks. Our motivation for evaluating Checkpoint-GCG in this auditing setting is that training models to be robust against attacks is an inherently difficult task and does not provide formal robustness guarantees. Stronger-than-realistic threat models allow us to discover vulnerabilities that may otherwise appear only as zero-day attacks, which remain unknown or undetected until they are exploited.
>
> **Attacking.** In Section 5.3, we indeed also relax assumptions to evaluate Checkpoint-GCG beyond the auditing scenario. For the strongest model (SecAlign-defended Llama-3-8B-Instruct), we summarize how the ASR changes as we progressively relax assumptions:
> - **Auditing setup** (full input prompt access + intermediate checkpoints access): 88% ASR.
> - **Universal attack** against the same model (intermediate checkpoints access only): 75.3% ASR.
> - **Universal and transferable attack** against a different model (Llama-3.1-8B-Instruct defended with SecAlign++; no access to full input prompts or intermediate checkpoints): 78.3% ASR with white-box access to the target model; 63.9% ASR with black-box query-only access to the target model.
>
> From requiring access to both intermediate checkpoints and full input prompts, to just black-box query-only access, the ASR indeed drops but is still very high (above 60%). We believe this to be a meaningful result as a) one attack out of two would actually succeed and that b) both SecAlign [1] and SecAlign++ [2] report near-zero ASRs across all prior attacks they have evaluated.
>
> Given the rapid growth of open-source model releases and defenses, it is hard to know what typical deployment scenarios would look like. They may vary widely, and in some cases similar surrogate models may indeed be accessible. For example, SecAlign and SecAlign++ are both open-source defense recipes, and strong models defended with SecAlign++ are already released [3], making them plausible candidates for real-world deployment. We show that Checkpoint-GCG is an effective attack in such cases, constituting a meaningful threat for at least some real-world deployments on top of the applicability of Checkpoint-GCG for auditing.
>
>
> [1] Chen, Sizhe, et al. “SecAlign: Defending Against Prompt Injection with Preference Optimization”, ACM CCS 2025. Link: https://arxiv.org/abs/2410.05451
>
> [2] Chen, Sizhe, et al. “Meta SecAlign: A Secure Foundation LLM Against Prompt Injection Attacks”. Link: https://arxiv.org/abs/2507.02735
>
> [3] https://huggingface.co/facebook/Meta-SecAlign-8B

---

> > ### Author Response · Authors · 2025-11-22
> >
> > > (W2) Checkpoint-GCG runs GCG many times across selected checkpoints; while GRAD reduces cost, the totals in reported experiments are nontrivial (per-sample totals reported in Table 4). The method uses large per-checkpoint budgets (authors set T up to 1000 with early stopping), which may be computationally expensive in practice for large models.
> >
> > Thank you for raising this point. While the total number of Checkpoint-GCG steps is indeed non-trivial, we believe it to remain reasonable relative to prior work and standard GCG, especially in the auditing setup where the goal is to evaluate new defenses and the cost is small relative to defense development. Our new results also show that this cost can be further reduced and that there is a trade-off between the number of selected checkpoints and the per-checkpoint budget. We expand on these points below.
> >
> > Regarding total Checkpoint-GCG steps, Table 4 of the submission compares multiple checkpoint selection strategies with varying hyperparameters, including straightforward or sub-optimal baselines (e.g., FREQ), which naturally incur a high number of total steps. Indeed, GRAD is the selected approach as it achieves the highest ASRs while requiring the fewest total Checkpoint-GCG steps. In addition to the $\tau_\text{grad}$ values in Table 4, we further increased the $\tau_\text{grad}$ threshold and we report the additional results below.
> >
> > | Checkpoint Strategy | Parameter values | ASR % ↑ | # Selected checkpoints ↓ | Total Checkpoint-GCG steps (avg across samples) ↓ |
> > |--|--|--|--|--|
> > |grad|$\tau_{\text{grad}}=0.05$| 100 | 102 | 3077 |
> > ||$\tau_{\text{grad}}=0.1$ | 100 | 64 | 2033 |
> > ||$\tau_{\text{grad}}=0.2$ | 90 | 41 | 1764 |
> > ||$\tau_{\text{grad}}=0.3$ (new) | 90 | 29 | 1475 |
> > ||$\tau_{\text{grad}}=0.4$ (new) | 50 | 19 | 1721 |
> > ||$\tau_{\text{grad}}=0.5$ (new) | 45 | 12 | 1213|
> > ||$\tau_{\text{grad}}=0.6$ (new) | 20 | 5 | 798 |
> >
> >
> > These results show that although the total number of Checkpoint-GCG steps is typically 2-4x the standard GCG budget of $T=500$, these additional steps yield substantial improvements in attack effectiveness. With roughly 3x the standard GCG budget, Checkpoint-GCG achieves ASRs above 90%, whereas standard GCG achieves only 6% (Table 1). In an auditing setting and at least some high-risk attack settings (e.g., zero-day attacks, attacks against agentic payment systems), we believe this trade-off is justified: a modest increase in compute significantly improves the chances of discovering vulnerabilities that standard methods miss.
> >
> >
> > We also observe a trade-off between the number of selected checkpoints and the average number of GCG steps required at each checkpoint. Lowering $\tau_\text{grad}$ increases the number of selected checkpoints, but the number of per-checkpoint GCG steps decreases accordingly. As a result, the total Checkpoint-GCG cost does not increase substantially, while having more selected checkpoints allows for smoother optimization toward the final checkpoint and higher ASRs. We illustrate this trade-off in a new plot (Figure 3b) and discuss the results in Section 5.2 and Appendix C.
> >
> >
> > Regarding the per-checkpoint budget, we agree that $T=1000$ (with early stopping) is indeed a high per-checkpoint budget. We chose this value in our evaluation as a conservative upper bound. In reality, however, Checkpoint-GCG requires far fewer steps. The new Figure 3b shows that for SecAlign-defended Llama-3-8B-Instruct, the most robust model we evaluated, Checkpoint-GCG took around 30 steps per attacked checkpoint on average to achieve a 100% ASR.

---

> > > ### Author Response · Authors · 2025-11-22
> > >
> > > > (W3) The paper lacks systematic interpretability experiments or mechanistic explanations for why Checkpoint-GCG achieves superior attack performance.
> > >
> > > Thank you for the suggestion! Figure 2 in the paper provides a first intuition for why Checkpoint-GCG achieves superior performance, however, we agree that more insights would be valuable. We thus have performed an additional experiment investigating how the model’s *attention patterns* shift with Checkpoint-GCG suffixes.
> > >
> > > Inspired by recent work that analyzes changes in model activations and attention patterns to understand and detect prompt injection attacks [1,2], we now analyze how the model’s attention patterns shift with Checkpoint-GCG suffixes.
> > >
> > > Specifically, we compute the final model’s attention score of user prompt ($Attn_{U}$) and its attention score of injected instruction + suffix ($Attn_{A}$), with suffixes obtained at each checkpoint during Checkpoint-GCG. We plot these attention scores across all samples where Checkpoint-GCG against the final model is successful in Figure 5a (see Section 5.4 for experimental details).
> > >
> > > We find that these suffixes, obtained during Checkpoint-GCG’s optimization, smoothly and monotonously shift the final model’s attention away from the user prompt towards the adversarial injection. Although these suffixes are optimized against intermediate checkpoints and not the final model, they still progressively steer the final model’s attention in the right direction – indicating that the greedy optimization of Checkpoint-GCG at each checkpoint gradually steers the suffix towards a successful attack. Notably, these attention scores evolve in a remarkably smooth and monotonous manner, especially as Checkpoint-GCG does not directly optimize for attention scores. We then also show that, for checkpoints where these attention scores shift the most, Checkpoint-GCG spends most of its budget (Figure 5b).
> > >
> > > Taken together, we really appreciate your suggestion, and believe this experiment and its insights further improve our work. We have now added this experiment and discussion to Section 5.4.
> > >
> > > [1] Hung, Kuo-Han, et al. "Attention tracker: Detecting prompt injection attacks in LLMs" Findings of the Association for Computational Linguistics: NAACL 2025. 2025. Link: https://arxiv.org/abs/2411.00348
> > >
> > > [2] Abdelnabi, Sahar, et al. “Get my drift? Catching LLM Task Drift with Activation Deltas”, 2025 IEEE Conference on Secure and Trustworthy Machine Learning (SaTML). Link: https://arxiv.org/abs/2406.00799
> > >
> > > **We hope our responses and additional experiments adequately address your concerns and provide further clarity. If so, we would be grateful if you might consider updating your assessment of the paper.**

---

### Author Response · Authors · 2025-12-03
**Final Remarks**

We sincerely thank all reviewers for their valuable feedback and suggestions, which helped us improve the paper. As the discussion phase comes to an end, we here provide an overview of our contributions, new experiments added during the rebuttal, and why we believe our work offers value to the community.

In this work, we propose Checkpoint-GCG, a method that leverages intermediate model checkpoints as stepping stones to guide the GCG attack towards successful adversarial suffixes against increasingly robust models. Specifically:
- We propose Checkpoint-GCG as a method to **audit the robustness of state-of-the-art, fine-tuning-based defenses against prompt injection attacks**. We show Checkpoint-GCG to find successful attacks in 88% of the cases against the strongest defended model evaluated, while regular GCG only finds 6%. This shows that successful attacks do exist against SOTA fine-tuning-based defenses, revealed by a stronger auditor, and that weaker attacks like GCG become ineffective for auditing as defenses improve.
- We show that **universal attacks** exist against SOTA prompt injection defenses. The universality of the attack is indeed crucial for prompt injections to be a risk in practice, as neither the system prompt, nor the user task, for which the injected context is retrieved is known in advance. Standard GCG does indeed sometimes succeed at finding non-universal attacks, but it fails to find universal ones.
- We introduce four **checkpoint selection strategies** and extensively evaluate each under varying hyperparameters. Our results show how model training dynamics can be leveraged to improve attack effectiveness and efficiency.
- While our focus is on model auditing, we also show our attack to **transfer black-box** across models of the same family even if they are fine-tuned with a different prompt injection defense, providing important implications for model developers.

Following the feedback and suggestions from the reviewers, we have strengthened our work during the rebuttal phase by adding:
- **Interpretability experiments** showing that Checkpoint-GCG suffixes steadily shift the final model’s attention away from the user prompt and toward the injection (Section 5.4).
- Additional results for **checkpoint selection strategies**, showing that total GCG steps can be further reduced while maintaining high ASR (Table 4). We now also show the trade-off between the number of selected checkpoints, GCG steps per checkpoint, and ASR in Figure 3b.
- **Theoretical intuition** grounded in the defenses’ fine-tuning objective and Checkpoint-GCG’s optimization objective (Appendix B).
- Preliminary experiments showing that suffixes discovered by Checkpoint-GCG during the auditing process can be **incorporated back into the fine-tuning pipeline to strengthen defenses**, whereas incorporating suffixes found by standard GCG yields little improvement (Appendix H).
- We furthermore sincerely appreciate reviewer 62dP for pointing out the paper by Wang et al. [1], which proposes jailbreaking an aligned model by attacking a sequence of checkpoints from easy to hard. We have now added an appropriate citation (Section 6), and implemented their binary search strategy for selecting checkpoints, showing it to take roughly 3 times as many GCG steps as our approach (Appendix C).
- Comparison of the distribution of the total GCG steps for Checkpoint-GCG vs. standard GCG, showing that 45% of Checkpoint-GCG’s successful attacks required <1000 total steps across all checkpoints (Appendix D).
- Evaluation of universal suffixes on another benchmark (CyberSecEval2) shows that these suffixes maintain high ASRs out-of-the-box on unseen samples (Section 5.3).

While progress has been made in defenses, prompt injection remains a “frontier, unresolved security problem” [2], and in security, “99% is a failing grade” [2,4] and motivated adversaries will spend significant time and resources to find successful attacks [3,4]. By assuming a stronger-than-realistic threat model in auditing, Checkpoint-GCG can help discover vulnerabilities **before** they are exploited as zero-day attacks. We believe the community will benefit not only from the proposed method as an auditing tool but also from the insights regarding defense mechanisms, interpretability, and the risks of universal and transferable prompt injection attacks.

[1] Wang, Zi, et al. “Functional Homotopy: Smoothing Discrete Optimization via Continuous Parameters for LLM Jailbreak Attacks”, ICLR 2025. Link: https://arxiv.org/abs/2410.04234

[2] https://simonwillison.net/2025/Oct/22/openai-ciso-on-atlas/

[3] https://openai.com/index/prompt-injections/

[4] https://simonwillison.net/2023/May/2/prompt-injection-explained/#prompt-injection.015

---

### Meta-Review · Area_Chair_zVgs · 2026-01-03

**Summary:**

This paper aims to audit and evaluate the robustness of fine-tuning based prompt injection defenses (such as StruQ and SecAlign). Compared with standard GCG, the idea of this paper is to leverage checkpoints of the fine-tuned LLMs. The experimental results demonstrate that the proposed method can achieve higher ASRs.

**Reviewer Concerns:**

In general, the reviewers think this paper is easy to follow, and the problem studied in this paper is important, i.e., evaluating the robustness of the state-of-the-art fine-tuning prompt injection defenses. I appreciate the efforts of the authors in addressing the comments from the reviewers. However, there are some concerns about the proposed methods. First, the checkpoints are not always available in practice, which limits the applicability of the proposed method. Second, the paper may discuss the technical novelty and compare the performance of the proposed work with existing studies. There are already many previous papers that refine GCG to improve the attack performance for jailbreak and prompt injection, including nano-GCG (an improved implementation of GCG).

**Reviewer Scores:**

The reviewers may not change their scores.

---

### Decision · Program_Chairs · 2026-01-26

Reject